

# Microbial communities associated with the black morel *Morchella sextelata* cultivated in greenhouses

Gian Maria Niccolò Benucci[1,*], Reid Longley[2,*], Peng Zhang[3], Qi Zhao[3], Gregory Bonito[1,2] and Fuqiang Yu[3]

[1] Plant Soil and Microbial Sciences, Michigan State University, East Lansing, MI, USA
[2] Microbiology and Molecular Genetics, Michigan State University, East Lansing, MI, USA
[3] CAS Key Laboratory for East Asia Biodiversity and Biogeography, Kunming Institute of Botany, Chinese Academy of Sciences, Yunnan, China
* These authors contributed equally to this work.

Corresponding authors
Gregory Bonito, bonito@msu.edu
Fuqiang Yu, fqyu@mail.kib.ac.cn

## ABSTRACT

Morels (*Morchella* spp.) are iconic edible mushrooms with a long history of human consumption. Some microbial taxa are hypothesized to be important in triggering the formation of morel primordia and development of fruiting bodies, thus, there is interest in the microbial ecology of these fungi. To identify and compare fungal and prokaryotic communities in soils where *Morchella sextelata* is cultivated in outdoor greenhouses, ITS and 16S rDNA high throughput amplicon sequencing and microbiome analyses were performed. *Pedobacter*, *Pseudomonas*, *Stenotrophomonas*, and *Flavobacterium* were found to comprise the core microbiome of *M. sextelata* ascocarps. These bacterial taxa were also abundant in the soil beneath growing fruiting bodies. A total of 29 bacterial taxa were found to be statistically associated to *Morchella* fruiting bodies. Bacterial community network analysis revealed high modularity with some 16S rDNA operational taxonomic unit clusters living in specialized fungal niches (e.g., pileus, stipe). Other fungi dominating the soil mycobiome beneath morels included *Morchella*, *Phialophora*, and *Mortierella*. This research informs understanding of microbial indicators and potential facilitators of *Morchella* ecology and fruiting body production.

## INTRODUCTION

Morels (*Morchella* spp.) are an iconic genus of edible mushrooms that are distributed across the Northern hemisphere (*O'Donnell et al., 2011*). Morels have a long history of use in Europe, and are sought after in North America and Asia. They remain an economically important culinary mushroom today, and are commercially harvested in the springtime when they fruit naturally (*Obst & Brown, 2000*; *Pilz et al., 2007*). For example, in western North America, morels have been estimated to contribute $5–10 million to the economy through direct sales (*Pilz et al., 2007*).

*Morchella* is a species-diverse genus. Classical taxonomic treatments of *Morchella* based on morphological characters are complicated by the extreme variation in macro-characters

(*Richard et al., 2015*). Recent efforts have reconstructed the phylogeny and biogeographic history of this genus with multiple genetic loci and have helped to stabilize the taxonomy of *Morchella* (*O'Donnell et al., 2011*; *Du et al., 2012*; *Richard et al., 2015*). From these studies, over 66 phylogenetic species of *Morchella* are recognized and shown to belong to three clades: the Elata clade (black morels), the Esculenta clade (yellow morels) and the Rufobrunnea clade (garden morels) (*Taskin et al., 2012*; *Richard et al., 2015*). Most morel species are confined geographically to particular continents and regions (*O'Donnell et al., 2011*), yet, a few species such as *Morchella rufobrunnea* and *M. importuna* appear to be more widely distributed, perhaps through recent human-mediated transport and long-distance dispersal (*Elliott et al., 2014*).

While attempts to cultivate morels have been ongoing for decades (*Costantin, 1936*), methods remained elusive until the 1980s, when protocols for cultivating morels indoors were devised and patented (*Ower, 1982*; *Ower, Mills & Malachowski, 1986*, *1989*). Recently, methods for cultivating black morels (Elata clade) in soils under greenhouse environments were developed in China, leading to a significant increase in morel production. Morels are cultivated in non-axenic soils by planting fertile spawn in the soil, and feeding the mycelium with exogenous nutrient bags once it emerges from the soil (*Guo et al., 2016*; *He et al., 2017*; *Liu et al., 2018a*). However, as with other agronomic crops there is variability in production, and problems with diseases may occur during production (*Guo et al., 2016*; *He et al., 2017*; *Liu et al., 2018a*). Bacteria are thought to be responsible for the promotion of primordia differentiation and ascocarp growth, and may help suppress diseases (*Liu et al., 2017*). Consequently, there is interest in understanding the microbial ecology of morels during their cultivation to improve production and to improve diseases detection and management.

A series of bipartite lab experiments indicate that the bacterium *Pseudomonas putida* can stimulate sclerotium formation of morel isolates (*Pion et al., 2013*). This association was demonstrated to benefit *Morchella*'s carbon status. A more recent study found that bacteria belonging to Proteobacteria, Chloroflexi, Bacteroides, Firmicutes, Actinobacteria, Acidobacteria, and Nitrospirae were associated with soils of outdoor morel cultivation systems (*Liu et al., 2017*). *Liu et al. (2017)* showed that the soil bacterial communities, as well as morel yields, were influenced by variations in trace elements such as Fe, Zn, Mn, and their complexes. At the genus level *Pseudomonas*, *Geobacter*, and *Rhodoplanes* were the most predominant detected overall, with *Pseudomonas* having the highest abundance in the control group, *Rhodoplanes* dominated in the single-element groups (Zn, Fe, and Mn) and *Geobacter* were lower in the control group than in most experimental groups.

Consequently, it was hypothesized that distinct bacterial consortia associated with morel growth stage and fruiting bodies would be detected. It is expected that this would include *Pseudomonas*, which has been found to be a beneficial associate of morels previously (*Pion et al., 2013*), as well as other taxa (*Pion et al., 2013*). It is also hypothesized that fungal pathogenic lineages may be detected, since greenhouses were dominated by a single cultivated species (*M. sextellata*). To test these hypotheses, high throughput amplicon sequencing was used to assess fungal (ITS rDNA) and prokaryotic (16S rDNA) communities from an outdoor morel cultivation environment. This study provides

in-depth characterizations of fungal and prokaryotic communities associated with *M. sextelata* and the soils beneath their fruiting bodies.

## METHODS

### Sampling microbial communities associated with morel fruiting bodies and soils beneath fruiting bodies

Morel fruiting bodies and soils beneath growing morels were sampled from a high-tunnel greenhouse in Caohaizi Village, Xundian County, Kunming City, Yunnan Province, China, where the black morel *M. sextelata* was being cultivated. The site is situated 1,950 m in elevation. The pileus and stipe from five mature (>10 cm) and five immature (<1 cm) fruiting bodies were sampled by placing a piece of tissue roughly one cm$^2$ in size into CTAB 4X buffer with a flame sterilized razor. Approximately two cm$^3$ of soil was also sampled from directly below each morel fruiting body. Soils were dried completely with silica beads and were kept on silica until processing (described below). In total, microbial analyses were performed on 20 samples, 10 *M. sextelata* ascocarps (five young and five mature), and 10 soils beneath the ascocarps, which were analyzed for both bacterial (16S rDNA) and fungal (ITS rDNA) communities. Bacterial communities were determined for 10 morel ascocarps, including pileus ($n = 10$, five mature and five immature) and stipe ($n = 10$, five mature and five immature) tissues.

### Molecular methods

DNA was extracted from ~0.5 g of dried and homogenized soils with the PowerMag® Soil DNA Isolation Kit (Qiagen, Carlsbad, CA, USA) following manufacturer's recommendations. Morel tissues were ground with a sterile micro pestle and then extracted using standard chloroform extraction protocol (*Trappe, Trappe & Bonito, 2010*). Extracted DNA was amplified using DreamTaq Green DNA Polymerase (ThermoFisher Scientific, Waltham, MA, USA) with the following primer sets: ITS1f-ITS4 for Fungi and 515F-806R for Bacteria and Archaea, following a protocol based upon the use of frameshift primers as described by *Chen et al. (2018)* and *Lundberg et al. (2013)*. PCR products were stained with ethidium bromide, separated through gel electrophoresis, and imaged under UV light. Amplicon concentrations were normalized with the SequalPrep Normalization Plate Kit (ThermoFisher Scientific, Waltham, MA, USA) and pooled. Amplicons were then concentrated 20:1 with Amicon Ultra 0.5 mL 50K filters (EMD Millipore, Darmstadt, Germany) and purified with Agencourt AMPure XP magnetic beads (Beckman Coulter, Brea, CA, USA). A synthetic mock community with 12 taxa and four negative (no DNA added) controls was included to assess sequencing quality (*Palmer et al., 2018*). Amplicons were then sequenced on an Illumina MiSeq analyzer using the v3 600 cycles kit (Illumina, San Diego, CA, USA). Sequence reads have been submitted to NCBI SRA archive under the accession number PRJNA510627.

### Bioinformatic analyses

Sequence quality was evaluated for raw forward and reverse Illumina ITS and 16S reads with FastQC (http://www.bioinformatics.babraham.ac.uk/projects/fastqc/). Selected reads

were demultiplexed in QIIME according to sample barcodes (*Caporaso et al., 2010*). Forward reads were then cleaned from the Illumina adapters and sequencing primers with Cutadapt (*Martin, 2011*), quality filtered, trimmed to equal length (*Edgar & Flyvbjerg, 2015*; *Edgar, 2016*), de-replicated, removed from singleton sequences and clustered into operational taxonomic units (OTUs) based on 97% similarity following the UPARSE algorithm (*Edgar, 2013*). Taxonomy assignments were performed in QIIME with the RDP Naïve Bayesian Classifier (*Wang et al., 2007*) using the Greengenes database (*DeSantis et al., 2006*) version gg_13_8 for 16S rDNA, and with CONSTAX (*Gdanetz et al., 2017*) based on the UNITE fungal ITS rDNA sequence database version 7.1 2016-08-22 (*Kõljalg et al., 2005*) (Fig. S1).

## Statistical analyses

The *otu_table.biom* (*McDonald et al., 2012*) with embedded taxonomy classifications and *metadata.txt* files for each marker gene were imported into the R statistical environment for analysis (*R Core Team, 2018*). Before proceeding with the analysis, data were quality filtered to remove OTUs with less than 10 total sequences (*Lindahl et al., 2013*; *Oliver, Callaham & Jumpponen, 2015*). OTUs that appeared in the negative controls (i.e., contaminants) were removed across all samples when ≥10 reads were present in any single control. Observed OTU richness (*S*) (*Simpson, 1949*), Shannon's diversity index (*Hill, 1973*), and Evenness (*Kindt & Coe, 2005*) were used as α-diversity metrics. The Shannon index (*H*) was calculated as $H = -\sum_{i=1}^{R} p_i \ln p_i$ where $p_i$ the proportion of individuals belonging to the $i$ species in the dataset, while the OTU evenness (E) was calculated as $E = \frac{H}{ln(s)}$ where $H$ is the Shannon diversity index and $S$ the observed OTU richness. Diversity indexes were with the "specnumber" and "diversity" functions in R package *vegan* (*Oksanen et al., 2019*) and with the function "diversityresult" in the package *BiodiversityR* (*Kindt & Coe, 2005*). After assessing for data normality and homogeneity of variances significant differences between mean alpha-diversity measures were found with ANOVA and Tukey's tests. Rarefaction curves were used to assess OTU richness from the results of sampling (Figs. S2 and S3). To avoid biases and data loss in some groups of samples due to inherent variations in alpha-diversity in soils compared to morels, OTUs were normalized using the R package *metagenomeSeq* before calculating β-diversity (*Paulson et al., 2013*). Principal coordinate analysis (PCoA) was used to investigate community β-diversity with the function "ordinate" from the *phyloseq* package (*McMurdie & Holmes, 2013*). Diversity patterns were then tested for statistical differences across sites in the *vegan* R package with the PERMANOVA function "adonis" and tested for homogeneity of variances with the function "betadisper." OTUs that showed high and significant correlation with sample groups were identified through the function "multipatt" in the *indicspecies* package (*De Cáceres & Legendre, 2009*).

To assess co-occurrences among OTUs a bipartite network was produced for the prokaryotic communities with the "spiec.easi" function in the *SpiecEasi* R package (*Kurtz et al., 2015*). To build the network, the following parameters were used: lambda.min. ratio=1e-2, nlambda=50, rep.num=99. The network was constructed using the OTUs present in at least 15 samples to increase the sensitivity of the analysis. After assessing

network stability using the "getStability" function in *SpiecEasi*, general (i.e., modularity, sparsity, transitivity) and individual OTUs (i.e., degree, closeness centrality, betweenness centrality, articulation points) network indexes were calculated. The network was visualized with the Fruchterman-Reingold layout with $10^4$ permutations as implemented in the *igraph* R package (*Csardi & Nepusz, 2006*). A heatmap showing abundances of prokaryotic OTUs statistically associated with *Morchella* ascocarps was created using the *ComplexHeatmap* R package (*Gu, Eils & Schlesner, 2016*). All statistical analyses and graphs were performed in R version 3.4.4 (*R Development Core Team, 2018*).

# RESULTS

## High-throughput sequencing results

After quality filtering, a total of 215,201 reads were analyzed with an average read depth of 21,520 across 10 samples for the ITS marker and 2,237,810 reads with an average read depth of 74,593 reads across 30 samples for 16S rDNA. After removing contaminants, as well as negative and mock samples, a total of 509 OTUs for fungal communities and 5,169 OTUs for prokaryotic communities were obtained. Our synthetic mock community matched the 12 artificial taxa, which were sequenced alongside with the samples. No mock sequences were detected in any other libraries indicating that barcode switching was not an issue in this study.

## Fungal and prokaryotic communities composition of *Morchella sextelata* fruiting bodies and associated soils

The fungal communities in soils beneeth *Morchella* fruiting bodies were dominated overall by Ascomycota (72.9%), Mucoromycota (7.1%), and Basidiomycota (3.4%). The fungal communities of the substrate beneath the young *Morchella* ascocarps were dominated by *Morchella* sp. (39.0%), *Phialophora* sp. (15.6%), and *Mortierella* (8.7%). Under the mature *Morchella* ascocarps, the same most abundant taxa were detected, but with different relative abundances: *Morchella* sp. (58.2%), *Phialophora* sp. (15.6%), *Mortierella* (5.3%). Relative abundances at family level (Relative abundance >1%) for each analyzed sample are also reported in the barplot (Fig. 1A).

Differences in community composition associated with pileus, stipe, or soil niches were detected in 16S rDNA communities. A barplot of relative abundances at phylum level (relative abundance >1%) of the prokaryotic communities are shown in Fig. 1B. The whole prokaryotic community was dominated by Bacteroidetes (36.7%), Proteobacteria (23.7%), and Actinobacteria (12.3%). The prokaryotic communities in the pileus of *Morchella* ascocarps were dominated by Bacteroidetes (53.3%) and Proteobacteria (43.9%). The most abundant genera were *Pedobacter* (38.7%), *Pseudomonas* (28.3%), and *Flavobacterium* (10.6%). In the stipe of *Morchella* ascocarps the dominant phyla were Bacteroidetes (89.2%) and Proteobacteria (9.2%), which included the genera *Pedobacter* (83.1%), *Flavobacterium* (4.9%), and *Pseudomonas* (2.4%). In the soil beneath *Morchella* ascocarps the dominant prokaryotic phyla were Actinobacteria (26.1%), Chloroflexi (19.8%), and Proteobacteria (19.8%). The most abundant genera were an uncultured bacterium in the

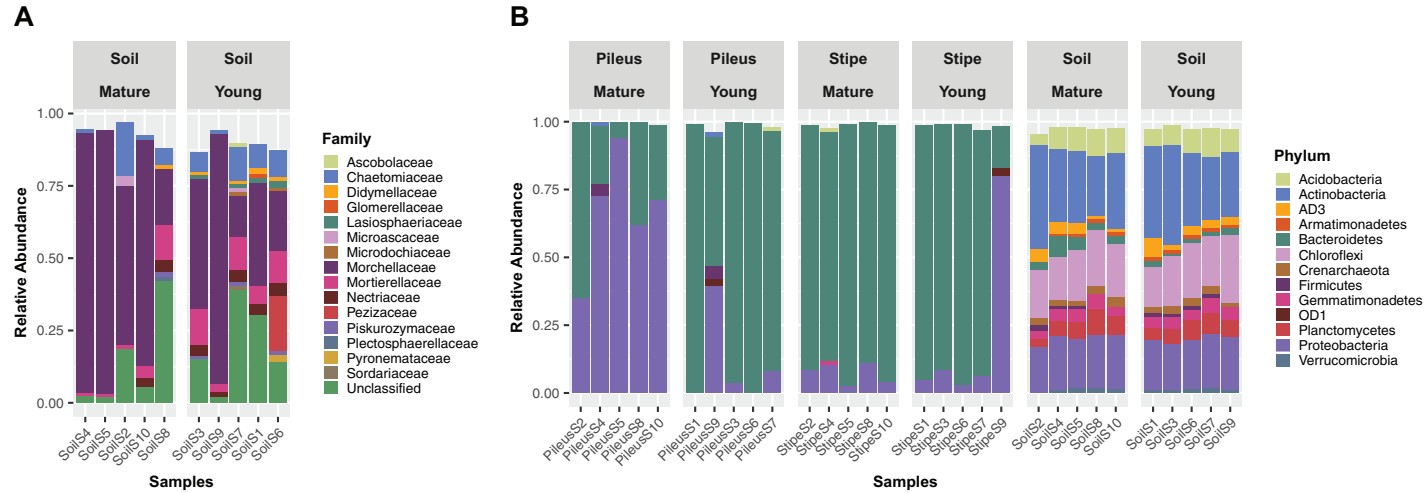

**Figure 1 Stacked bar plots.** Stacked bar plots showing fungal families (A) with relative abundance ≥1% detected in soil beneath ascocarps of mature and young *Morchella sextelata* fruiting bodies, and prokaryotic phyla (B) with relative abundance ≥1% detected in pileus, stipe, and soils beneath ascocarps of mature and young *M. sextelata*.

Gaiellaceae (6.6%), an uncultured bacterium in the Ellin6529 clade (6.1%) and *Kaistobacter* (3.5%) (Fig. 1B).

## Microbial richness and evenness in soils beneath *Morchella* fruiting bodies

Significant differences ($p \leq 0.05$) in OTU richness of the prokaryotic community were found between soil, stipe, and pileus samples (Table 1). The soil compartment showed almost 10-fold higher richness than was present in *Morchella* pileus or stipe compartments. Similar trends were true for both Evenness (E) and Shannon index (H) diversity measurements. No differences were found when average alpha-community measures were compared between young and mature morel samples. Fungal alpha diversity trended to be slightly higher in the samples of the young *Morchella*, but this was not statistically significant (Table 1).

## Fungal and prokaryotic community β-diversity in *Morchella* samples

Principal coordinate analysis (PCoA) ordination graphs performed on the 16S rDNA data show that the difference between the soil from pileus and stipe prokaryotic communities explained the variance obtained in the first axis (49.9%), while differences between pileus and stipe samples are evident in the second axis (18.3%) (Fig. 2A). PCoA ordination graphs performed on ITS soil data show that the variance of the first axis (66.3%) is due to differences between samples collected under mature compared to young *Morchella* fruiting bodies (Fig. 2B). Variation obtained for the second axis (8.8%) is due to the high heterogeneity (See below) of the samples collected under young *M. sextelata* fruiting bodies. PERMANOVA analysis of the 16S dataset show that there was a significant effect of the maturity stage of *Morchella* samples on the prokaryotic communities (Table 2).

**Table 1 Mean OTU richness (S), Evenness (E), and Shannon diversity index (H) detected in the prokaryotic and fungal communities.**

|  |  | Pileus | Stipe | Soil |
|---|---|---|---|---|
| Prokaryotes | Richness (S) | 245.30 ± 74.67[a] | 310.80 ± 61.28[a] | 3231.20 ± 221.92[b] |
|  | Evenness (E) | 0.23 ± 0.05[a] | 0.20 ± 0.04[a] | 0.80 ± 0.01[b] |
|  | Shannon (H) | 1.26 ± 0.30[a] | 1.10 ± 0.19[a] | 6.44 ± 0.05[b] |
|  |  | **Mature** | **Young** |  |
|  | Richness (S) | 1218.33 ± 338.26 | 1306.53 ± 388.63 |  |
|  | Evenness (E) | 0.42 ± 0.08 | 0.40 ± 0.08 |  |
|  | Shannon (H) | 2.96 ± 0.67 | 2.90 ± 0.70 |  |
|  |  | **Mature** | **Young** |  |
| Fungi | Richness (S) | 205.40 ± 37.85 | 284.6 ± 31.51 |  |
|  | Evenness (E) | 0.28 ± 0.07 | 0.5 ± 0.08 |  |
|  | Shannon (H) | 1.52 ± 0.45 | 2.87 ± 0.50 |  |

**Note:**
Different letters represent statistically significant differences (Tukey test after ANOVA, $p \leq 0.05$).

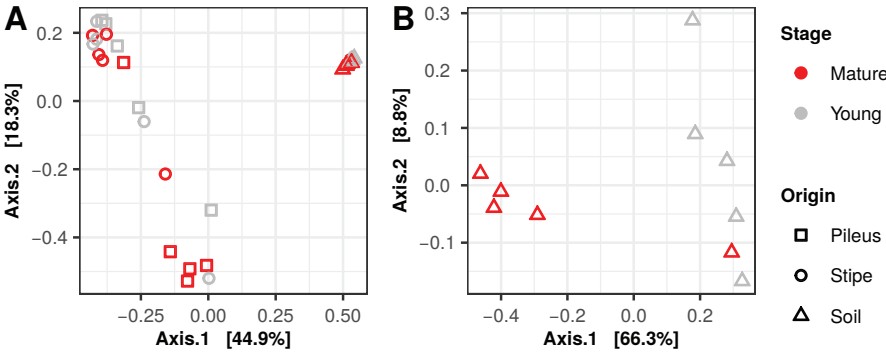

**Figure 2 Principal coordinates analysis plots, using Bray–Curtis dissimilarity matrices, of prokaryotic (A) and fungal (B) communities associated with *Morchella sextelata*.**

The PERMANOVA analysis of the ITS dataset show that there was a significant effect of maturity stage of *M. sextelata* fruiting bodies on the soil fungal communities (Table 2) that was not due to sample group dispersion (Fig. S5).

## Indicator species and intersections between stage and site

Several prokaryotic OTUs were significantly associated with the pileus, stipe, or associated soil (and combination of them) portions of *M. sextelata* fruiting bodies (Table S1). A heatmap of the OTUs associated with *Morchella* fruiting bodies (i.e., associated to pileus, stipe, stipe and pileus, soil and pileus, stipe and soil) is provided in Fig. 3. Two OTUs were statistically associated to *Morchella*'s pileus: *Corynebacterium* sp. and *Pseudanabaena* sp. Two OTUs were also associated to the *Morchella* stipe: *Granulicatella* sp. and an unidentified OTU in Coxiellaceae. All other OTUs reported in the heatmap were associated to two different groups. Among these OTUs, one specific *Pedobacter* sp.1 was
**Table 2 Permutational multivariate analysis of variance (*adonis*) and multivariate homogeneity of groups dispersions analysis (*betadisper*) results for both prokaryotic and fungal communities associated with *Morchella* soil and fruiting bodies.**

| | Factor | PERMANOVA | | | | DISPERSION | |
|---|---|---|---|---|---|---|---|
| | | D*f* | *F*-value | $R^2$ | *p*-value | *F*-value | *p*-value |
| Prokaryotes | Stage | 1 | 1.156 | 0.022 | 0.297 | 0.618 | 0.438 |
| | Origin | 2 | 12.651 | 0.471 | **0.001** | 9.627 | **<0.001** |
| | Stage:Origin | 2 | 1.655 | 0.062 | 0.112 | | |
| | Residuals | 24 | | | | | |
| | Total | 29 | | | | | |
| Fungi | Stage | 1 | 0.698 | 0.432 | **0.027** | 0.011 | 0.917 |
| | Residuals | 8 | | | | | |
| | Total | 9 | | | | | |

**Note:**
Significant *p*-values at $p \leq 0.05$ are highlighted in bold.

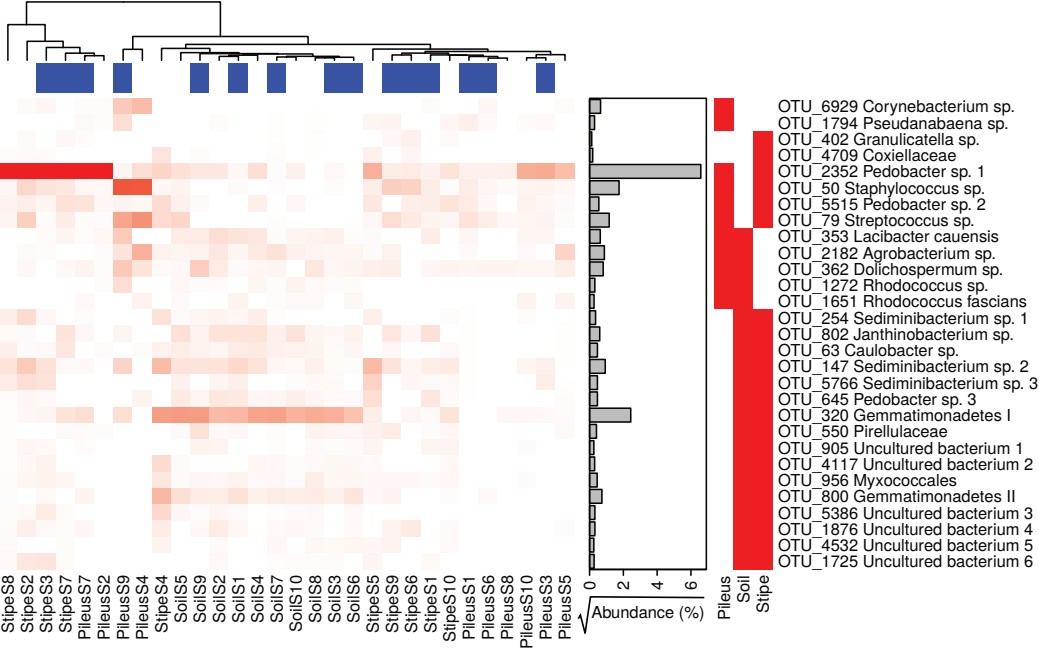

**Figure 3 Heatmap of the relative abundances of the 29 indicator taxa significantly associated with *Morchella sextelata* pileus, stipe, pileus and stipe, pileus and soil, stipe and soil.** Samples are ranked according the clustering dendrogram. Blue and white blocks of the top annotation represent samples from young and mature morels, respectively. The side annotation barplot reports the square root of the cumulative relative abundance for each OTU across all the samples.

associated to both pileus and stipe and was more abundant in these two compartments than it was in the soils.

Venn diagrams show that soil samples contained a high number of unique prokaryotic OTUs (3,239) compared to pileus (63), and stipe (34) samples (Fig. 4A) in contrast to what

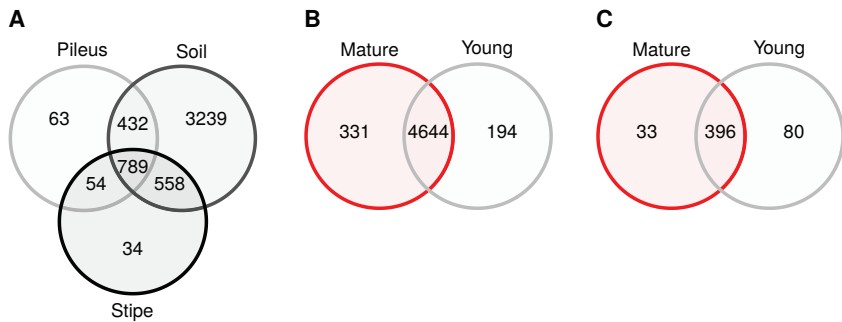

**Figure 4 Venn diagrams showing core and unique OTUs among different sample groups.** (A) Prokaryotic communities in pileus, stipe, and soils beneath *Morchella sextelata*; (B) Prokaryotic communities in mature and young ascocarps of *M. sextelata*; (C) Fungal communities in mature and young *M. sextelata* ascocarps.

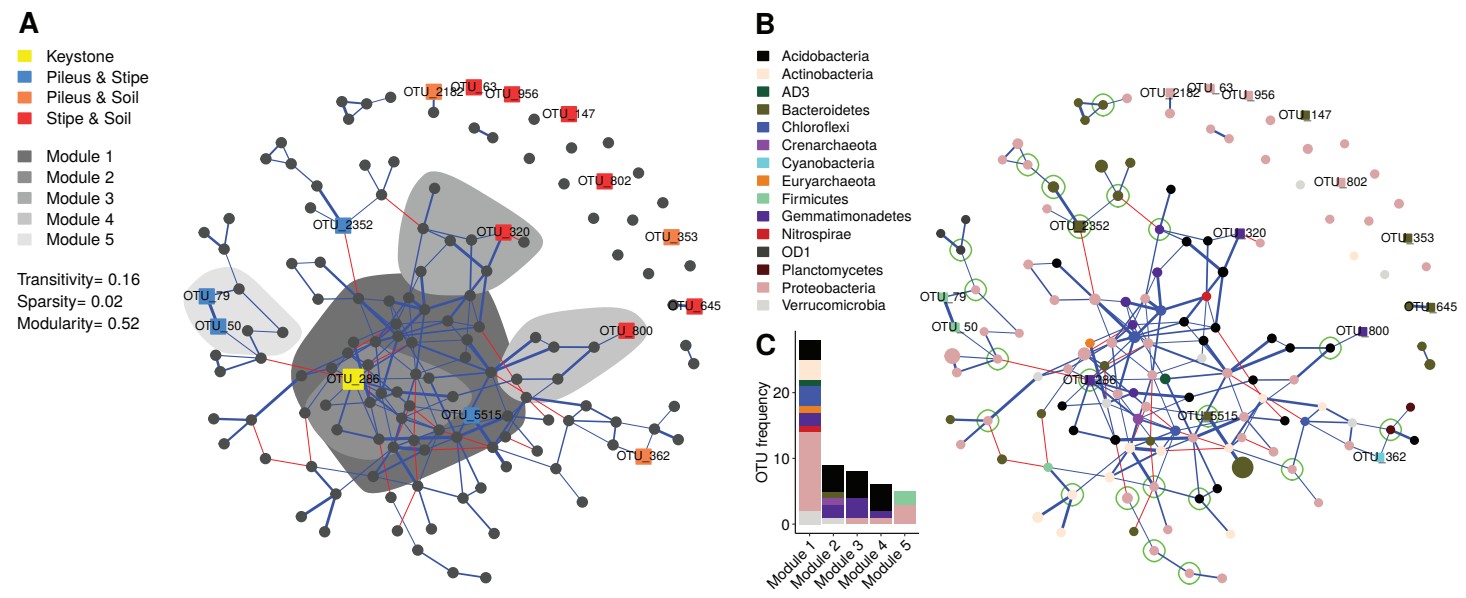

**Figure 5 Microbial co-occurrence network showing the prokaryotic community structure of *Morchella sextelata*.** Each node (vertex) indicates a single OTU at 97% sequence similarity. Blue edges indicates positive co-occurrence, red edges indicated negative co-occurrences; (A) Network showing indicator species (see in Fig. 3), keystone OTU, and the first top five modules. (B) Network showing the taxonomic composition of each node and articulation points. Nodes size is the square root of the relative OTU abundance; (C) Barplot showing OTU frequency (OTU richness) and taxonomic composition for the first five modules.

was shared among them (789). Most bacterial OTUs detected in *Morchella* fruiting bodies were found in young and mature fruiting bodies (4,644), with only 194 and 331 uniquely present in young or mature samples, respectively. In the fungal communities, mature and young morel soils shared 396 OTUs, while 33 and 80 OTUs were only present in mature or young specimen, respectively (Fig. 4C).

## Network analysis

The bipartite network (140 vertex, 199 edges, stability = 0.044) that was obtained is a sparse network (Figs. 5A and 5B), having a low number of possible edges (sparsity ≈ 2%).

The network showed low transitivity (≈0.15) which is a measure of the probability that the adjacent vertices of a vertex are connected. The network showed high modularity (≈0.5) which measures the division into subgraphs (i.e., communities or modules) in which vertex (i.e., OTUs) are more interconnected together than with the rest of the network. A total of 45 modules were identified, with the first five modules containing 40% of the total OTUs: Module 1 contained 29 OTUs; Module 2 contained nine OTUs; Module 3 contained eight OTUs; Module 4 contained six OTUs; Module 5 contained five OTUs. A total of 17 modules were composed of one single OTU (Fig. 5A). Several modules were peripheral and negatively connected (edge weight max = 0.34, min = −0.20) to other modules. Module 5 contained two indicator OTUs identifies for the pileus and stipe niche. Most of the indicator taxa for the stipe and soil environments were in single OTU modules (see Fig. 3 for taxonomic position), disconnected from the main network. Taxonomic classifications at phylum level of each OTU in the network is shown in Fig. 5B. Proteobacteria, Acidobacteria, and Gemmatimonadetes were dominant in the first five modules (Fig. 5C). Interestingly, archaeal OTUs in the Euryarchaeota, Crenarchaeota were also present in the network. In addition to identifying nodes with high degree (number of connections), some OTUs were identified as articulation points, node whose removal disconnects the network (e.g., OTU_2352).

## DISCUSSION

Black morels are cultivated in greenhouse conditions in non-sterilized soils (Liu et al., 2018a). It has been hypothesized that fungi and bacteria living in these substrates may facilitate, or conversely, inhibit developmental transitions and fruiting body development (Liu et al., 2017). Soils where morels are cultivated successfully were highly colonized by Morchella mycelium, especially in soils beneath mature morel fruiting bodies. The morel mycelium inoculated in soils appears to overgrow and potentially exclude other fungal taxa.

Regarding prokaryotic communities, Pedobacter, Pseudomonas, Stenotrophomonas, and Flavobacterium were dominant in the microbiome of M. sextelata fruiting bodies. The high abundance of Pseudomonas (Proteobacteria) in morel fruiting bodies raises questions concerning their roles in the development of morels, following observations on the occurrence and diversity of bacterial communities on Tuber magnatum during truffle maturation, Pseudomonas putida farming by M. crassipes (Pion et al., 2013), and the abundance of Pseudomonas OTUs in soils where black morels are cultivated in Sichuan, China (Liu et al., 2017).

Strong effects of fungal host identity have been seen on the structure of bacterial communities in other mushroom species (Pent, Põldmaa & Bahram, 2017). Interestingly, Pseudomonas, Flavobacterium, Janthinobacterium, and Polaramonas were also detected in fruiting bodies of Pezizales truffle species through 16S rDNA surveys of the fruiting bodies (Benucci & Bonito, 2016; Splivallo et al., 2019), including, Kalapuya brunnea, which belongs to the Morchellaceae family (Trappe, Trappe & Bonito, 2010). Selective filtering of bacterial communities by the fungal host has also been shown for other fungi, such as Tuber (Barbieri et al., 2005, 2007; Antony-Babu et al., 2014; Splivallo et al., 2015, 2019;

*Benucci & Bonito, 2016*; *Amicucci et al., 2018*), *Cantharellus* (*Kumari, Sudhakara Reddy & Upadhyay, 2013*; *Pent, Põldmaa & Bahram, 2017*), *Tricholoma* (*Oh et al., 2018*), *Agaricus* (*Rossouw & Korsten, 2017*; *Aslani, Harighi & Abdollahzadeh, 2018*), *Suillus, Leccinum, Amanita, and Lactarius* (*Pent, Põldmaa & Bahram, 2017*; *Liu et al., 2018b*). As reported in Table 3, Proteobacteria are some of the most abundant bacterial genera associated with fruiting bodies of different fungal lineages based on recent published literature.

Moreover, the relative abundances of bacterial groups varied between vegetative (stipe) and fertile (pileus) tissues of morel mushrooms, as well as from the soil beneath them. For instance, the pileus of *Morchella* was enriched in *Pseudomonas, Stenotrophomonas*, and *Flavobacteria* compared to stipe microbial communities. The stipe was mostly colonized by *Pedobacter* (83%) compared to the pileus (39%) and the soil where it accounted for only 0.4% of relative abundance of bacteria. OTUs classified as *Pedobacter* were statistically associated to pileus and stipe tissues and were present in different modules in the microbial network. This indicates that the pileus tissue may recruit a specific set of prokaryotic taxa which are not recruited to the stipe. This is supported by a significant reduction in prokaryotic richness in the pileus and stipe compared to the soils. Of interest, the two tissue types also smelled different. Previous studies have indicated differences in the chemical composition of *Amanita* pileus and stipes due to metabolite production in the fruiting body (*Deja et al., 2014*). If similar chemical differences exist between *Morchella* pileus and stipe, this could offer an explanation for the existence of different prokaryotic communities within distinct tissues of the *Morchella* fruiting body and the soil beneath them.

*Morchella* pileus, stipes, and soils were also shown to be specific niches for other indicator bacterial taxa. Surprisingly, human and animal (sometimes plant) pathogens such as *Corynebacterium, Granulicatella, Streptococcus*, and *Staphylococcus* were found exclusively associated to the pileus and/or stipe environment (*Collins et al., 2004*; *Cargill et al., 2012*). These taxa are components of the microbial network associated with *Morchella* fruiting bodies (Fig. 5), although they were found in peripheral modules that were negatively connected with the main structure. Some other taxa such as *Lacibacter* (*Qu et al., 2009*) or *Sediminibacterium* (*Qu & Yuan, 2008*), which are bacteria common in soil, were also identified as indicator species but were not present in our network.

It has been hypothesized that microbes in the soil are necessary for morel fruiting to occur. The role of *Pseudomonas* in the cultivation of button mushrooms (*Agaricus bisporus*) has been studied previously, and was shown to increase both yield and primordia formation (*Zarenejad, Yakhchali & Rasooli, 2012*; *Chen et al., 2013*; *Pent, Põldmaa & Bahram, 2017*). The relative abundance of *Pseudomonas* species increased throughout cultivation cycle of *Agaricus bisporus* and peaked around the time of fruiting (*Chen et al., 2013*). It was also shown that the presence of specific strains of *Pseudomonas putida* in *Agaricus* inoculum increased mushroom yields by as much as 14% (*Zarenejad, Yakhchali & Rasooli, 2012*). Previous research found that *Pseudomonas putida* stimulates sclerotia formation in *Morchella* (*Pion et al., 2013*). These results are consistent with our findings that *Pseudomonas* are abundant in soils and fruiting bodies of cultivated morels, thus, they may be important in the growth and fruiting of these fungi. *Liu et al. (2017)*

**Table 3 List of the top abundant bacterial genera associated to fungal fruiting bodies of different fungal taxa found in this study and from the literature.**

| Family | Fungal species | Bacterial genera | Isolation method | Origin | Reference |
|---|---|---|---|---|---|
| Agaricaceae | *Agaricus bisporus* | *Microbacterium, Pseudomonas, Ewingella, Enterobacter* | Culture dependent | Pileus/ Stipe | *Aslani, Harighi & Abdollahzadeh (2018), Rossouw & Korsten (2017)* |
| Amanitaceae | *Amanita* spp. | *Pseudomonas, Janthinobacterium, Enterobacter, Burkholderia, Acinetobacter* | Culture independent | Pileus/ Stipe | *Pent, Põldmaa & Bahram (2017), Liu et al. (2017)* |
| Boletaceae | *Leccinum* spp. | *Burkholderia, Chryseobacterium, Novosphingobium* | Culture independent | Pileus/ Stipe | *Pent, Põldmaa & Bahram (2017)* |
| Chantarellaceae | *Chantarellus* spp. | *Chitinophaga, Rhizobium, Bacteroides, Hafnia, Enterobacter* | Culture independent | Pileus/ Stipe | *Pent, Põldmaa & Bahram (2017), Kumari, Sudhakara Reddy & Upadhyay (2013)* |
| Morchellaceae | *Morchella sextelata* | *Pedobacter, Pseudomonas, Stenotrophomonas, Flavobacterium* | Culture independent | Pileus/ Stipe | This study |
| | *Leucangium carthusianum* | *Pseudomonas, Jantinobacterium* | Culture independent | Gleba | *Benucci & Bonito (2016)* |
| | *Kalapuya brunnea* | *Jantinobacterium, Flavobacterium, Rhizobium, Pseudomonas* | Culture independent | Gleba | *Benucci & Bonito (2016)* |
| Russulaceae | *Lactarius rufus* | *Burkholderia, Shewanella, Dyella* | Culture independent | Pileus/ Stipe | *Pent, Põldmaa & Bahram (2017)* |
| Suillaceae | *Suillus bovinus* | *Burkoholderia, Corynebacterium, Pseudomonas* | Culture independent | Pileus/ Stipe | *Pent, Põldmaa & Bahram (2017)* |
| Tuberaceae | *Tuber borchii* | *Sinorhizobium/Ensifer, Bradyrhizobium, Rhizobium, Microbacterium* | Culture dependent | Gleba | *Barbieri et al. (2005), Splivallo et al. (2015)* |
| | *Tuber aestivum* | *Bradyrhizobium, Polaromonas, Pseudomonas* | Culture independent | Gleba | *Splivallo et al. (2019)* |
| | *Tuber magnatum* | *Sinorhizobium, Bradyrhizobium, Rhizobium, Variovorax* | Culture dependent | Gleba | *Amicucci et al. (2018), Barbieri et al. (2007)* |
| | *Tuber melanosporum* | *Bradyrhizobium, Polaromonas, Variovorax, Propionibacterium* | Culture independent | Gleba | *Antony-Babu et al. (2014), Benucci & Bonito (2016)* |
| Tricholomataceae | *Tricholoma matsutake* | *Pseudomonas, Serratia, Mycetocola, Ewingella, Stenotrophomonas* | Culture dependent | Pileus/ Stipe | *Oh et al. (2018)* |

also demonstrated that *Pseudomonas* are the most common bacteria overall in soils where morels are cultivated, with the highest abundance in the treatment having the highest yield of morel ascocarps, however, bacterial associated with morel fruiting bodies was not assessed. The effect of *Flavobacterium* spp. on mushroom fruiting body formation is not well studied, but these bacteria have been shown to be associated with the successful cultivation of *Pleurotus ostreatus* (*Cho et al., 2008*). Thus, it is possible that *Flavobacterium* contribute to the formation of mushroom fruiting bodies.

The recruitment of prokaryotic communities by *Morchella* may occur due to a selection by the fungus for specific taxa, or because it offers a preferential niche for bacterial growth. It is also possible that these two factors act simultaneously. For example, *Cantharellus cibarius* is populated by millions of different bacteria that are thought to be existing on fungal exudates including trehalose and mannitol (*Rangel-Castro, Danell & Pfeffer, 2002*). Fast growing bacteria that live on fungal-derived nutrients may occupy this niche quickly and may play a role in inhibiting the entry of other bacteria or pathogens (*Liu et al., 2018b*). Future studies can directly test these hypotheses by assessing the importance of management and specific bacterial taxa on the morel microbiome and fruiting body production.

## CONCLUSIONS

In conclusion, our work adds further evidence that the fungal host plays a role in the selective recruitment of specific bacterial taxa. Our study found that the *Morchella* microbiome is consistently comprised of a small community of bacteria, including *Pedobacter*, *Pseudomonas*, *Stenotrophomonas*, and *Flavobacteria*, which appear to be recruited from the soil and enriched in fungal fruiting body tissues. Among those, *Pedobacter* was enriched in and significantly associated with the pileus environment in respect to the stipe and soil compartments. Although some of the bacteria groups detected on morels have also been detected in other mushrooms, based on this preliminary study, many microbial taxa may be exclusive to *Morchella*. The role of host identity may provide predictive explanation for differences between microbiomes of morels and other mushrooms. Future research is warranted to test the function of these bacteria on morel fruitification and management.

## ACKNOWLEDGEMENTS

The authors are grateful to Caohaizi Village for use of facilities and allowing and assisting us with sampling in this study. The authors confirm they have no conflicts of interest pertaining to this research.

### Funding

This work was supported by Michigan State University AgBioResearch NIFA project MICL02416 and Project GREEEN GR17-083 for Gregory Bonito. Gregory Bonito and Fuqiang Yu were financially supported by the Science and Technology Service Network Initiative, Chinese Academy of Sciences (2017) and Guizhou Science and Technology

Program project number 4002 (2018). Reid Longley graduate research fellowship support from the Plant Biotechnology for Health and Sustainability Training Program Project NIH T32-GM110523. The funders had no role in study design, data collection and analysis, decision to publish, or preparation of the manuscript.

## Grant Disclosures

The following grant information was disclosed by the authors:
Michigan State University AgBioResearch NIFA: MICL02416 and GREEEN GR17-083.
Science and Technology Service Network Initiative, Chinese Academy of Sciences (2017).
Guizhou Science and Technology Program: 4002 (2018).
Plant Biotechnology for Health and Sustainability Training Program Project NIH T32-GM110523.

## Competing Interests

The authors declare that they have no competing interests.

## Author Contributions

- Gian Maria Niccolò Benucci analyzed the data, prepared figures and/or tables, authored or reviewed drafts of the paper, approved the final draft.
- Reid Longley analyzed the data, prepared figures and/or tables, authored or reviewed drafts of the paper, approved the final draft.
- Peng Zhang performed the experiments, approved the final draft.
- Qi Zhao conceived and designed the experiments, performed the experiments, approved the final draft.
- Gregory Bonito conceived and designed the experiments, performed the experiments, contributed reagents/materials/analysis tools, authored or reviewed drafts of the paper, approved the final draft.
- Fuqiang Yu conceived and designed the experiments, performed the experiments, contributed reagents/materials/analysis tools, authored or reviewed drafts of the paper, approved the final draft.

## Data Availability

   Data is available at the SRA: PRJNA510627.
   Code is available in GitHub: https://github.com/Gian77/Scientific-Papers-R-Code/tree/master/Benucci_etal_2019_MorchellaMicrobiome.

## Supplemental Information

Supplemental information for this article can be found online at http://dx.doi.org/10.7717/peerj.7744#supplemental-information.

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
