# Peer review of "Microbial communities associated with the black morel Morchella sextelata cultivated in greenhouses"

_PeerJ, doi:10.7717/peerj.7744_

## Round 0.1 · original submission · Minor Revisions

Overall, the reviews are quite positive. However, the authors should revise the manuscript following the reviewers' comments. In particular, please carefully address the comments related to the proper use of scientific terminology.

Reviewer 1 ·

Basic reporting

In some cases The MS does not meet all the required standard of the Journal from editing point of view.
1. Line numbers are missing.
2. The MS font is not unified. It is written in tow mixed fonts along the MS (Ariel and Ariel Unicode).
3. The citations within the MS are not written in the same font, some with italics and some are not.
4. The scientific names should be marked (italics) also in the list of references.
5. There is mixed used between stem and strip terms describing the same organ (stipe in text and stem in figures). Should use the same word.
6. Figure 3: What are represented by stem 7,3,2,8 and Stem 5,9,6,1,10 and the two groups of cups? Are they stems and cups of young and matured mushrooms? While the tow stem groups seem to be different in their prokaryotic profiles in the figure, there is no indication clarifying in text what they represent.
7. Fig 1:You have mentioned in the legend that fungal results are of fruiting bodies while in text you describe it as soil substrate samples.
8. Reference list: the references of Masaphy,S, 2010 and of Volk TJ, Leonard TJ. 1989 are found in the reference list, but are not mentioned in text. Please add them as citation in text.

Experimental design

No comments

Validity of the findings

No comments

·

Basic reporting

No comment

Experimental design

No comment

Validity of the findings

No comment

Additional comments

This study reported the microbiomes associated with the growing and fruiting of Morchella sextelata. The experiments and bioinformatic analyses were nicely conducted, and presented with enough details. The findings are novel, and add important knowledge into the environmental mechanisms triggering morel fruiting.

The manuscript is well written, but could be improved to a better quality by polishing some details in the wording and figures (see below). The order of some sentences could be adjusted to make the language more fluent. Basically it could be accepted after a few minor revisions.

Title:
“Morel mushroom” sounds a little weird. “Morel” means Morchella mushroom, so “morel mushroom” is like “chicken animal”, sounds redundant. A single word “morel” or “morels” is enough.

Abstract and main text:
Avoid to use too much “we”, “we found”, “we use”, etc. Passive sentences are better for describing the methods and results. Same to the main text. “Growhouse” should be “greenhouse”?

Pileus or cap, stipe or stem, should be used uniformly all through the article. Some figures used “cap” and “stem” while other places used “cap” and “stipe”. These are suggested to be uniformly corrected to “pileus” and “stipe”.

I polished the abstract as an example:

Morels (Morchella spp.) are iconic edible mushrooms with a long history of human consumption. During the fruiting course of morels, some microbial taxa are believed to be important for triggering the formation of primorida and development of fruiting bodies. Thus, there is interest in ecology of microbiomes associated with the morel. In order to identify and compare fungal and prokaryotic communities presented in the soil substrates locate in outdoor greenhouses where Morchella sextelata is cultivated, metabarcoding surveys of ITS and 16S amplicons are used. The results show that Pedobacter, Pseudomonas, Stenotrophomonas, and Flavobacterium comprise the core microbiome of M. sextelata. These bacterial taxa were also abundant in substrates beneath growing fruiting bodies. Twenty-nine bacterial taxa were statistically associated to Morchella fruiting bodies. The prokaryotic community network analysis revealed high modularity with OTU clusters such as (???, ???, ???) living in specialized niches (e.g. pileus, stipe). The soil mycobiome beneath morels was dominated by Morchella, Phialophora, and Mortierella. This research improved our understanding of microbial indicators and potential facilitators of Morchella fruiting body production.

Figures:

The figures are nicely drawn but the tags and labels could be polished to facilitate the readers understanding precisely. "Mature" to "Mature fruiting bodies". "Young" to "Young fruiting bodies". "Cap" to "pileus".
Captions of Fig. 1,3,4,5, and Table 2, are incomplete. These errors need correction.

Reviewer 3 ·

Basic reporting

no comment

Experimental design

The sampling method needs more detail. I suggest that you improve the description at line 100-108 to make the procedure repeatable.

For comparison, i think the soil sample with no fruiting bodies should be collected and analyzed.

Validity of the findings

In line 353, you concluded that " many microbial taxa appear to be unique to morchella". I would say that if the result obtained from the samples collected at more sites ( only one site in this work ) could be more convincing.

Additional comments

Recent years, morels are widely cultivated in non- axenic soils in China. The results of this work add the evidence that morel plays a role in the selective inrichment of specific bacterial taxa, which is important to understand the soil microbial niches for the morel growth. It also has practical implications that specific bacteria could be used as promotor for morel production. Because the results that you obtained were from only one sampling site, i suggest that the title of this paper should be " preliminary study of microbial communities associated with morel (morchella sextelata) mushrooms cultivated outdoors.".

Reviewer 4 ·

Basic reporting

As for the language, there are no comments. It is suggested only to check if "cooccurrence" should be written all attached or not. In the text it is written in both ways (line 166, line 584,586).
In the abstract, line 33 it is not clearly described whether the findings are in the Morphella carpophore.
Furthermore, in the abstract it is not clearly explained whether the fungal populations were also searched for in the Morchella carpophores, since in the abstract we speak only of soil.
Regarding the references, they are certainly complete and up-to-date, nevertheless, important European works on the truffle are absent, line 293 and 294. For example Amicucci et al 2019, Barbieri et al 2007.
One observation: why in line 93 states that the authors hypothesize the presence of pathogenic bacteria? and why then in line 313 is it stated that "surprisingly" pathogenic fungi have been found? Explain this point better.
Regarding the structure of the work, there are small imperfections to be checked. In particular the font size is different. Check lines 144-146, 216, 259-265, 568-569, and also in the rest of the text.
The figures are fine, they are almost all clear, with some exceptions. The caption of figure 1, line 568-569, does not mention that 1B refers to bacteria. Obviously, looking at the diagram and the list of phylums you can guess. For clarity, the authors could add in the diagram "Fungal Family" in figure 1A and "Bacterial phylum" in 1B.
Still referring to figure 1, and to the results it exhibits, it is not explained either in the figure or in the text, why the authors decided not to analyze the fungal population in the different tissues of the fungus, as instead was done for the study of bacterial populations. Still referring to this experiment, it is not explained why the study in the different stages of maturation was not done for the bacterial analysis such as for fungal analysis.

Experimental design

The experimental design is well done and structured, even if the classification of microorganisms remains at the level of genus and not of species. Some small improvements can be made. In particular line 103-104, it is not described how the degrees of maturation of the samples under analysis were defined, nor what the degree of maturation to which the analysis took place. In the paragraph that describes the sampling, line 108, we suggest to define better what it refers to when writing that 40 samples have been analyzed in all.
We also suggest to write more clearly the procedure of concentration and preparation of amplicons, line 116 and later. Were the bands excised and purified by gel?
In line 117, explain why the 15% influence on the migration of ethidium bromide does not affect subsequent analyzes.
The conclusions are appropriate; a summary table would certainly be very interesting with a comparison of the microbial populations among the various fungal species analyzed in the literature, although not necessary for the purposes of publication.

Validity of the findings

no comment

---

## Round 0.2 · Minor Revisions

The manuscript has been improved by the authors. However, the reviewers still have a few minor comments.

Reviewer 1 ·

Basic reporting

Need more attention in regard to editing materials such as fonts and line spacing, addition of some data concerning the methodology, and little more in-depth discussion to compare this work to previous work on similar topic.

Experimental design

suggesing to write a little more in-depth discussion to compare this work to previous work on similar topic, i.e. on cultivated Morchella (Liu et al 2017).

Methods - need to add some small detailes

Validity of the findings

The findings are intersting and all data have beedb provided.
To my opinion, the autors need to give more larification regarding the different in the aim and in the results from previous work done on Morcheela (Liu et al 2017)

Additional comments

Generally, this version of the MS is much improved, and it obvious that the authors maid efforts to reply.
However, unfortunately, some of earlier comments were not treated.
Here are some of earlier comments and new ones. Hope this will improve the published MS.
My suggestion is that the MS will go through a professional paper editing to avoid these small writing errors.

Line numbers are according the word TrackChange file.

Old comments:
1. The MS font is still not unified. It is written in tow mixed fonts along the MS (Ariel and Ariel Unicode). See for example lines 171-174, 245-247 ect. Also line spacing is not unified.

2. Reference- the scientific names are still not marked (italics) in the list of references.

3. There is still use of the words we in many occasions along the MS. Please avoid the use of we, "we hypnotize" ect. It is suggested to use "it is hypnotize"


New comments:
1. Abstract: line 40, please add "beneath growing fruiting bodies" after the word soil, as you have not analyzed or any soil or substrate, but soil beneath growing fruiting bodies in particular.

2. Methods: Line 120- please add "beneath growing fruiting bodies" after the word "soil"
3. Methods: line 127- There are 30 samples and not 20.
4. Methods : Line 134- You have mentioned how the soil DNA was extracted " DNA was extracted from ~0.5 g of dried and homogenized soils with the PowerMag®" , but no indication of how the mushroom parts were treated, and DNA extracted.

5. Introduction and discussion: please indicate what the different between the current work and the work of Liu et al. 2017. Is it only the outdoor and greenhouse cultivation? Is it the mushroom species? Is it the mineral that are used? Are the mushrooms in the current work had any enrichments by minerals? There is no discussion on the overall differences between the origin of the samples and also not so much on the differences of the results if any?

·

Basic reporting

No comment

Experimental design

No comment

Validity of the findings

No comment

Additional comments

The authors have made satisfactory improvements to the manuscript. It can be accepted almost as it is, except that I still have two suggestions below. I trust that the authors can address the minor changes by themselves so that I don't request to see the revised manuscript again.
1. "Morels" usually means a community of morels consisting of different species. When referring to only Morchella sextelata, "a morel" is more accurate. Please revise this issue in the title as well as other places in the article.

2. Consider using "mycosphere soil" instead of "soil beneath the fruiting bodies". Define "mycosphere soil" as "soil beneath the fruiting bodies" at the first place it appears (and citation, doi: 10.1093/femsre/fuy008, Bacterial–fungal interactions: ecology, mechanisms and challenges), then use "mycosphere soil" afterwards althrough the article.

---

## Round 0.3 · accepted · Accept

The reviewers' concerns have been addressed, and the manuscript can be accepted for publication.